# The Emerging Role of Radiation Therapy in Renal Cell Carcinoma

**DOI:** 10.3390/cancers14194693

**Published:** 2022-09-27

**Authors:** Michael Christensen, Raquibul Hannan

**Affiliations:** Department of Radiation Oncology, University of Texas Southwestern Medical Center, Dallas, TX 75235, USA

**Keywords:** SBRT, SAbR, radiation, oligometastasis, oligoprogression, IVC tumor thrombus

## Abstract

**Simple Summary:**

Stereotactic ablative radiation therapy (SAbR) is a safe and effective local therapy for renal cell cancer (RCC) with emerging and evolving indications. In this review we provide an overview of the evidence to support SAbR for RCC in a variety of clinical settings.

**Abstract:**

Advancements in radiation delivery technology have made it feasible to treat tumors with ablative radiation doses via stereotactic ablative radiation therapy (SAbR) at locations that were previously not possible. Renal cell cancer (RCC) was initially thought to be radioresistant, even considered toxic, in the era of conventional protracted course radiation. However, SAbR has been demonstrated to be safe and effective in providing local control to both primary and metastatic RCC by using ablative radiation doses. SAbR can be integrated with other local and systemic therapies to provide optimal management of RCC patients. We will discuss the rationale and available evidence for the integration and sequencing of SAbR with local and systemic therapies for RCC.

## 1. Introduction

Kidney cancer encompasses multiple histologic types and different disease processes with varied clinical courses. The most common is renal cell carcinoma (RCC), representing 80 to 85% of primary renal neoplasms, and originates from the renal cortex. There are multiple subtypes of RCC, which include clear cell (75 to 85%), papillary (10 to 15%), chromophobe (5 to 10%), and oncocytoma (3 to 5%) [1]. Of patients with RCC, approximately 25% present with regional and distant metastasis at the time of diagnosis [2]. For patients with RCC who initially present with localized disease, up to 40% ultimately develop metastatic disease [2,3]. RCC has the potential to spread by local invasion through the surrounding tissue, venous drainage, lymphatic spread, or hematogenous dissemination. Surgery is the primary treatment modality for management of localized kidney cancer, and historically, systemic therapy for metastatic RCC was limited to cytokine therapies, including high-dose interleukin-2 (IL-2) and interferon. In recent years, the new standard of care for metastatic RCC (mRCC) has become systemic therapy with immune checkpoint inhibitors (ICI), tyrosine kinase inhibitors (TKI), or a combination of the two [4,5,6,7,8].

In addition to surgery and systemic therapy, radiation therapy is a promising treatment modality for RCC, although it was traditionally thought to be radioresistant. A study published in 1996 examined radiosensitivity in vitro in multiple human cancer cell lines, showing that RCC was relatively radioresistant to conventionally fractionated radiation therapy [9]. Additionally, a clinical trial published in 1987 showed that adjuvant conventional radiation therapy for RCC provided no improvement in local recurrence with severe toxicities, including death [10]. However, when delivered with a higher dose-per-fraction, RCC has subsequently been shown to be radioresistant in numerous in vivo and in vitro studies [11,12]. In the pre-clinical setting, human renal cell cancer grafts growing in a nude mouse model showed effective tumor control when treated with 48 Gy in 3 fractions [11]. 

Stereotactic ablative body radiation (SAbR) is defined by the American Society of Therapeutic Radiology and Oncology guidelines as a “treatment method to deliver a high dose of radiation to the target, utilizing either a single dose or a small number of fractions with a high degree of precision within the body [13].” SAbR has a broad spectrum of potential indications for various tumor types and locations [14] and has been assessed prospectively in multiple studies as both effective and safe [15,16,17,18]. Clinical experience using SAbR or hypofractionated radiotherapy (HFRT) for both intracranial and extracranial RCC metastases showed excellent local control (LC) rates of 90 to 98% [19,20,21]. One retrospective series of 50 patients with metastatic RCC who were treated with SAbR to 162 lesions showed a 90% LC rate with minimum toxicity at a median follow-up time of 37 months [19]. Wang et al. published a retrospective review of 175 metastatic RCC lesions treated with SAbR with a one-year LC rate of 91% and a favorable safety profile [22]. Their analysis further revealed improved outcomes with a biologically effective dose (BED) greater than 115 Gy. 

Integration of SAbR with surgery and the emerging systemic therapy landscape for RCC will lead to optimal outcomes for kidney cancer patients. This article will discuss the available evidence on combining SAbR with local and systemic therapies (summarized in Table), highlighting the lack of data and opportunities for future clinical trial development. 

## 2. SAbR for Primary RCC

The standard curative treatment for primary RCC is surgery, however patient characteristics such as inoperability or tumor size may favor other local treatments or observation. Other local therapies for primary RCC such as radiofrequency and cryoablations are invasive procedures with limitations based on size and location of the tumor, unlike non-invasive SAbR. Among the earliest outcomes to show promising results of SAbR for primary RCC are several retrospective studies published in the mid-2000s [20,23,24]. The first prospective dose escalation trials of SAbR showed that doses >27 Gy in 3 fractions did not have any failures and reported an overall LC of 93.7%, with failures only happening in earlier cohorts that received <10 Gy per each treatment [25] and escalation to 48 Gy in 4 fractions was achieved without dose-limiting toxicities [26]. Interestingly, lesions treated with SAbR decreased in size, however the tumor enhancement did not change on contrast imaging, suggesting that while the tumor cells were killed, the vasculature in the lesion was not affected. Another phase 2 trial of 37 patients with localized RCC treated with SAbR reported a LC of 100% at a median follow up of 24 months [27]. These authors also reported 3% grade 3 toxicity with no grade 4 to 5 toxicities. 

The International Radiosurgery Oncology Consortium for Kidney (IROCK) published a pooled analysis of outcomes from nine institutions for 223 patients who had RCC treated with SAbR [28]. In this cohort, the four-year LC, overall survival (OS), and progression-free survival (PFS) were 97.8%, 70.7%, and 65.4%, respectively. Only three (1.3%) patients experienced grade 3/4 bowel toxicity and the mean reduction in estimated glomerular filtration rate (eGFR) was 5.5 mL/min. This study showed a trend of worse PFS and cancer-specific survival (CSS) with larger tumor size [28,29]. A meta-analysis of 372 patients with RCC treated with renal SAbR showed random-effect estimates for LC of 97.2% and grade 3 to 4 toxicity of 1.5% [30]. A separate pooled analysis published in 2020 reaffirmed these positive results of SAbR for primary RCC [31]. The authors describe 95 patients deemed not suitable for surgery with primary tumors greater than 4 cm who were treated with definitive SAbR. This was effective with a four-year LC of 98.1%, no grade 3 to 5 toxicities, and impacted renal function with an average eGFR decrease of 7.9 mL/min.

## 3. Locally Advanced RCC

Standard of care treatment for patients with locally advanced RCC has traditionally been radical or partial nephrectomy. Postoperative treatment options have included observation and systemic therapy. More recently, investigators are exploring an increasingly nuanced approach given a variety of patient factors and outcomes.

Patients with newly diagnosed RCC have disease that invades the inferior vena cava (IVC) in up to 10% of cases. Disease invasion can extend from the renal vein and travel to the right atrium, with the extent of IVC disease portending a poor prognosis. If left untreated, IVC involvement can lead to venous congestion, Budd-Chiari syndrome, pulmonary embolism, or metastasis. Curative treatment for locally advanced RCC involving IVC tumor thrombus has traditionally only included surgery, however there is a 35% rate of high-grade perioperative morbidity, and 13% rate of peri- or post-operative mortality [32]. The surgical complications and subsequent risks of morbidity and mortality increases significantly for IVC tumor thrombi that goes past the hepatic vein. Despite the risks associated with this operation, an increased risk of relapse and metastasis still exists, including a greater than 40% one-year recurrence rate [33]. There are multiple possible mechanisms of this high rate of recurrence. The IVC tumor thrombus may invade the IVC wall, resulting in positive surgical margins leading to local recurrence. Alternatively, the IVC tumor thrombus may produce tumor emboli, thus causing metastatic spread.

SAbR may have application in the management of the RCC IVC tumor thrombus. An early case report of two patients treated with preoperative SAbR showed a median survival of 20 months at the time of publication, and no acute or late treatment-related toxicity [32]. A 15 patient retrospective, multi-institutional study reported a 58% response rate of RCC IVC tumor thrombus after SAbR, with symptomatic palliation in all patients and only grade 1 to 2 toxicity [34]. There are numerous potential indications for SAbR of IVC tumor thrombus, such as: palliation of Budd-Chiari syndrome; unresectable or recurrent disease after surgery; disease refractory to surgery and systemic therapy; cytoreduction with systemic therapy to increase resectability by alleviation of Budd-Chiari/hepatic venous congestion, which significantly improves surgical mortality; preoperative MRI evidence of IVC wall invasion; and patient eligibility for radical nephrectomy, but not tumor thrombectomy. Given the higher incidence of RCC in elderly and comorbid patients, it is conceivable that there will be many patients with RCC IVC tumor thrombus who would be candidates for the much less risky radical nephrectomy, but not a high-risk vascular surgery of tumor thrombectomy, particularly when cardio-pulmonary bypass may be required. In this setting, a reasonable option may be to treat the IVC tumor thrombus with SAbR combined with a simple radical nephrectomy. Another advantage of this strategy is that it can be performed in the local and community hospitals, whereas IVC tumor thrombectomies are usually limited to high-volume tertiary academic centers. 

Preoperative SAbR to the RCC IVC tumor thrombus is currently being investigated to reduce the high risk of recurrence. A safety lead-in phase II clinical trial of neoadjuvant SAbR for RCC IVC tumor thrombus (NCT02473536) is ongoing (Figure 1). The safety lead-in phase of the trial demonstrated that this treatment approach is feasible and safe, however, the oncologic outcome data is not yet complete [35]. This paradigm continues to evolve, and prospective evidence is currently lacking.

Consolidative or debulking SAbR may have applications in additional clinical scenarios. For instance, patients with locally advanced RCC without tumor thrombus may also be unresectable due to the extent of disease, medical inoperability, surgical risks, or simply due to a lack of evidence of clinical benefit as demonstrated by multiple clinical trials [36]. Singh et al. published feasibility data from a Phase 1 study that treated large kidney tumors with neoadjuvant SAbR [37]. Furthermore, patients initially diagnosed with metastatic disease may achieve a near complete response with systemic therapy, with only the primary tumor remaining. This could provide another scenario where SAbR can be utilized. There are ongoing multi-center phase 2 clinical trials (CYTOSHRINK NCT04090710 and SAMURAI NCT05327686) evaluating this strategy, leveraging potential synergy of SAbR with immunotherapy.

## 4. Oligometastatic RCC

Metastatic RCC represents an array of disease aggressiveness. Patients with International Metastatic Database Consortium (IMDC) low-risk disease may have a smoldering progression over many years, while patients with high-risk disease may die from their cancer in less than one year [38,39]. Additionally, metastatic RCC includes the range of patients with few involved sites to those with widely disseminated disease. Oligometastatic RCC can be divided into subcategories based on the risk of distant micrometastasis. These subcategories can be helpful to assess the probability of future progression at distant sites and the speed of progression.

The first subcategory are those patients that present with metachronous metastases more than a year after resection of the primary kidney tumor. These patients portend the best prognosis with an indolent disease. Glandular metastasis from RCC has also been reported to portend an indolent disease biology [40]. Treatment options for these patients include active surveillance, metastasectomy, SAbR, or systemic therapy [39,41,42,43]. The second subcategory of oligometastatic RCC are patients with favorable or intermediate IMDC risk. The risk of distant micrometastasis is sufficiently high so that systemic therapy will eventually be needed for these patients. However, upfront sequential SAbR can preserve health-related quality of life, as well as postpone available systemic therapy options. Retrospective and prospective studies of these patients have both shown that sequential SAbR can provide disease control for more than 15 months [42,43,44]. The third subcategory of patients with oligometastatic RCC are those with a high chance of distant micrometastatic disease with aggressive disease biology that is expected to rapidly progress, including those with IMDC high-risk, grade 4 histology, or sarcomatoid component histology. Up-front systemic therapy is necessary for these patients. However, while data is currently lacking, there may be a role for consolidation with SAbR to the oligometastatic metastatic sites with the rationale for debulking, eliminating the larger therapy-resistant metastasis or potential synergy with immunotherapy. 

Active surveillance is one treatment approach for appropriately selected indolent patients with oligometastatic RCC. A prospective trial of patients with oligometastatic RCC with proven indolent growth of metastases after primary nephrectomy showed that this subset of patients could safely undergo active surveillance for a median of 14.9 months before starting systemic therapy [39]. 

Metastasectomy is also a treatment option for patients with oligometastatic RCC, however data on surgical LC and safety are lacking [41]. A Japanese retrospective study of 1463 patients in which 20.8% underwent metastasectomy reported prognostic factors for metastatic RCC, including performance status, Hgb, LHD, serum calcium, C-reactive protein, and time from initial visit to metastasis being less than one year. Median survival for patients with no risk factors and one to two risk factors was 55.3 months and 29.6 months, respectively (one-year OS 92.8% and 76.6%, respectively) [45]. More recently, Tosco et al. investigated the survival impact of prognostic factors in patients with metastatic RCC who underwent metastasectomy [46]. Their results indicated that advanced primary tumor stage, high tumor grade, non-pulmonary metastases, disease-free interval of less than 12 months, and multi-organ metastases were independent factors for survival. Patients with 0 to 1, 2, 3, or greater than 4 factors had two-year cancer-specific survival rates of 95.8%, 89.9%, 65.6%, and 24.7%, respectively [46]. These tools may help clinical decision making for appropriate patient selection for local therapy. 

SAbR is a promising local treatment option that has not only shown favorable LC rates of greater than 90%, but can also provide an option for local therapy at an otherwise inoperable location. A phase II prospective trial from Sweden used SAbR in primary and metastatic RCC and showed an OS of 32 months with a LC rate of 79% at a median follow-up of 52 months [20]. Additionally, the University of Chicago published a prospective study with 81% of initial metastatic progression in patients with oligometastatic RCC limited to less than five sites after treatment with SAbR. This study also confirmed that approximately half of patients had either no or limited metastatic progression after a median follow up of 20.9 months [47]. Aggressive upfront sequential SAbR is supported by these experiences as an effective local therapy that can potentially control disease progression in patients with limited metastases. Retrospective analyses have supported the use of SAbR for oligometastatic disease to defer the start of systemic therapy and possibly extend survival [42]. This has recently become the subject of prospective studies, including one that supported the efficacy and safety of this approach with SAbR [48]. Moreover, this strategy can provide durable disease control in the setting of additional oligometastatic lesions with sequential, subsequent focal SAbR. This approach was described in a retrospective study where 30% of patients received two or more courses of SAbR to additional sites of metastatic disease [42]. The first phase II trial demonstrating the efficacy of sequential SAbR in the control of systemic therapy naïve oligometastatic RCC reported a one-year progression-free interval of 82.6%, and a one-year freedom from systemic therapy of 91.3% with no measured decline in patient-reported quality of life [44,49]. Tang et al. published a prospective feasibility phase II study where subsequent sites of progression were allowed to be treated with SAbR and found a median PFS of 22.7 months with acceptable toxicity [43]. While the study met its feasibility endpoint, it did not meet its estimated efficacy endpoint of 71% one-year PFS and reported a one-year PFS of 64%. 

While the safety of SAbR has been excellent in published series, caution must be exercised in certain scenarios. For instance, due to the vascular nature of RCC, ultra-central lung metastasis have rarely been shown to be associated with serious life-threatening late effects such as hemoptysis or hemothorax years after treatment. It is often difficult in these situations to assess the contribution of radiation, tumor recurrence, and systemic therapy as the etiology of the hemoptysis. A second potential cautionary scenario is the accentuation of future systemic therapy toxicities (i.e., colitis or pneumonitis), or a radiation-recall-type side effect. 

## 5. Oligoprogressive RCC

Individuals with metastatic RCC can develop disease progression at only a few select sites of disease, deemed oligoprogressive (Figure 2). To date, there has been limited research on patterns of progression. For example, conventionally used criteria for response assessment in clinical trials, such as Response Evaluation Criteria in Solid Tumors (RECIST) criteria, do not distinguish patterns of progression. The current approach to progression in clinical practice is to switch systemic therapy, even if progression is limited to a few sites. This is also an approach for patients who are otherwise tolerating the ongoing systemic therapy well. There are likely different modes of progression that reflect a spectrum of disease responsiveness to therapy and cancer biology. One hypothesis to explain limited disease progression while on systemic therapy is mutational heterogeneity and clonally propagated branched evolution that fosters tumor adaptation and therapeutic failure through Darwinian selection [50,51,52]. This spectrum of disease progression may thus be optimally managed with a strategy different from the singular standard approach to change systemic therapy. 

The introduction of focal therapies for controlling oligoprogressive sites (Figure 2) could be advantageous by increasing duration of the current therapy and preserving the limited available subsequent therapies. By extending duration of the current systemic therapy and altering the course of the disease through elimination of resistant metastasis, it is possible that this approach could also improve survival outcomes. In addition, reported subsequent lines of systemic therapy are typically associated with shorter PFS intervals and are often associated with increased toxicity [53]. Local therapy seems unlikely to undermine future systemic therapy, and such an approach may extend patient survival.

Multiple retrospective studies have evaluated SAbR for mRCC, however few papers have been published on the role of oligoprogression [19,54,55,56,57,58,59,60]. A retrospective analysis from Santini et al. included 55 mRCC patients on first-line systemic therapy with oligoprogression who were managed with focal approaches (including SAbR in 46%) and showed an mPFS of 14 months [55]. Another single-institution retrospective review of 72 patients with mRCC on systemic therapy treated with SAbR to oligoprogressive sites showed similar PFS, regardless of systemic therapy [60]. A multi-institutional study by Meyer et al. reported 180 patients with mRCC who had been treated with SAbR; of these, 101 patients were treated for oligoprogressive disease [56]. The median local recurrence-free survival, PFS, time to systemic therapy, and OS were 19.3, 8.6, 10.5, and 23.2 months, respectively. UT Southwestern Medical Center performed a retrospective review of SAbR for oligoprogression in mRCC, which showed a median mPFS of 9.2 months [54]. Data on this topic is emerging, with two prospective phase 2 trials showing that SAbR to oligoprogressive sites is able to extend the duration of ongoing systemic therapy [61,62]. The Canadian multi-center study showed that the change of TKIs can be delayed by more than a year with SAbR in this setting as a secondary outcome [61]. A second phase 2 trial demonstrated an extension of TKI or immunotherapy by a median of 11.1 months, with a median duration of SAbR-aided systemic therapy of 24.4 months [62]. Interestingly, patients on immunotherapy had a longer duration of PFS (12 months vs. 8 months; *p* = 0.04) compared to TKI, indicating a potential interaction of SAbR with immunotherapy. While multiple ongoing phase 2 trials are investigating this paradigm (NCT04974671; NCT04299646), larger phase 3 trials are needed in this space to confirm survival benefit of this strategy for oligoprogressive RCC patients. 

SAbR for oligoprogressive mRCC has been shown to be generally well-tolerated. Toxicity may also be exacerbated, however, by both ICIs and TKIs, and the safety of SAbR in conjunction with systemic therapy continues to be evaluated. SAbR with concurrent ICI/TKI was started with caution due to concerns for potential increased toxicity, but no enhanced toxicity was observed in initial studies [63,64,65]. This warrants the need for further prospective studies in this space. Mohamad et al. evaluated the safety of concurrent ICI and hypofractionated radiotherapy in 59 patients with mRCC, and concluded that adverse events of any grade did not significantly differ from historical rates of ICI therapy alone [66]. In a phase I trial, Tang et al. treated 55 patients with ipilimumab and either concurrent or sequential SAbR. They reported a 34% rate of grade 3 toxicity, comparable to treatment with ipilimumab alone [67]. Contrastingly, a meta-analysis of 13 prospective randomized trials with concurrent TKI and radiation therapy showed increased grade 3 or greater toxicity [68]. However, a different pooled analysis of 68 prospective trials of ICIs showed that those who received an ICI within 90 days following radiation therapy did not appear to be associated with an increased risk of serious adverse events [69].

## 6. CNS and Spine Metastasis

Brain metastases have been reported in up to 17% of patients with RCC [70]. With the recent approval of more effective systemic therapies, patients with mRCC live longer, and there is an expected increase in incidence of brain metastases for these patients [4,6,7,8]. Despite improvements in systemic therapies, the blood–brain barrier poses a persistent challenge to treat RCC brain metastases. This is why local therapy, such as surgery or radiation, remains a crucial treatment option for treating brain metastases [71]. Surgical resection has been a traditional treatment approach, however that may not always be possible due to patient or tumor factors. Classical radiation treatment for intracranial metastases has generally involved whole-brain radiation therapy (WBRT); however, this paradigm has shifted to prefer stereotactic radio surgery (SRS). SRS for RCC-specific brain metastases also allows greater dose-per fraction treatments to combat what has traditionally considered a radioresistant histology. SRS has less neurocognitive toxicity without a survival detriment compared to WBRT with SRS [72]. LC rates have been excellent, reaching as high as 98% to 100% in certain series [70,73,74,75,76]. SRS not only has significant clinical advantages, but there are logistical benefits as well. SRS is a minimally invasive outpatient procedure, it can be performed on patients unfit for surgery, and is feasible in certain intracranial locations deemed unresectable.

Second to pulmonary metastasis, osseous involvement is the next most common site of metastasis and can occur in up to 27% of patients with mRCC [77]. The spinal column is the most common site of osseous metastasis from RCC [78]. Treatment options include conservative pain management, steroids, surgery, radiotherapy, or a combination of these. A multidisciplinary approach is highly recommended for RCC spinal metastasis because great consideration is needed when deciding one therapy over another, and clinicians must consider specific clinical factors such as the severity of a patient’s pain, neurologic symptoms, presence of spinal cord compression, or associated edema [79]. Patients with RCC and isolated spine metastasis or otherwise oligometastatic disease may be considered for curative intent local therapy. SAbR, including single-fraction treatments, for RCC spine metastases has been shown to provide an 83% LC at one-year, few to no grade 3 or greater toxicity, as well as fast, durable pain relief [80,81]. 

Ablative radiation alone may not be feasible if the tumor is causing spinal cord compression or cord abutment, and a surgical decompression and debulking is performed followed by high-ablative radiation to achieve durable LC. One retrospective review showed that postoperative SAbR following epidural spinal cord decompression provided a one-year LC greater than 95% [82]. Another option is a multi-modal approach with neoadjuvant systemic therapy followed by local therapy when the metastasis has extensively infiltrated the spinal canal and the proximity of the spinal cord keeps from delivering an ablative radiation dose or safe surgical resection. Moreover, osseous metastases from RCC are lytic and can cause significant cortical destruction, placing patients at increased risk for compression fractures. SAbR can increase the risk of vertebral compression fracture further, and it is therefore recommended to pursue prophylactic kyphoplasty [83]. Surgical resection for RCC metastasis also poses an intraoperative bleeding risk that can be addressed with arterial embolization prior to resection. Consequentially, a multi-disciplinary approach is ideal for the proper management of spinal metastasis from RCC. 

## 7. Palliation

Aside from the previously outlined setting of CNS, there are various additional scenarios where SAbR may be indicated for the treatment of RCC with curative, consolidative, and adjuvant intent. Additionally, multiple indications for palliative irradiation for RCC also exist. The most common sites of metastatic disease in patients with RCC have been documented as: lung (45%), bone (30%), lymph node (22%), liver (20%), brain (9%), and adrenal (9%) [84]. Indications for palliative radiation include radiologic evidence of metastatic disease and a corresponding sign or symptom such as pain, spinal cord compression, superior vena cava syndrome, brain metastasis, fracture and prevention of fracture in the weight bearing bones, bleeding, and more. Hematuria is a frequent presenting symptom for metastatic RCC that can be palliated effectively with radiation therapy [85]. Due to the classic radio-resistance of RCC to conventional fractionation, hypofractionation schemes favoring a higher dose per fraction are preferred. The regimen of 20 Gy in 5 fractions is preferred over 30 Gy in 10 fractions. Whenever possible, applicable dose escalation should be considered with intensity-modulated radiation therapy or SAbR. 

## 8. Conclusions

SAbR is both an established and emerging treatment option with curative or palliative intent, ranging from early inoperable RCC to oligometastatic RCC to widely metastatic RCC. Given SAbR’s safety and efficacy for both primary and metastatic RCC, the onus is on the physician to successfully integrate this modality with the available and emerging local and systemic therapies in order to maximize outcomes for RCC patients. While several clinical trials are ongoing, many more are required to provide high-level prospective evidence regarding integration of SAbR for the management of primary and metastatic RCC (summarized in Table 1).

## Figures and Tables

**Figure 1 cancers-14-04693-f001:**
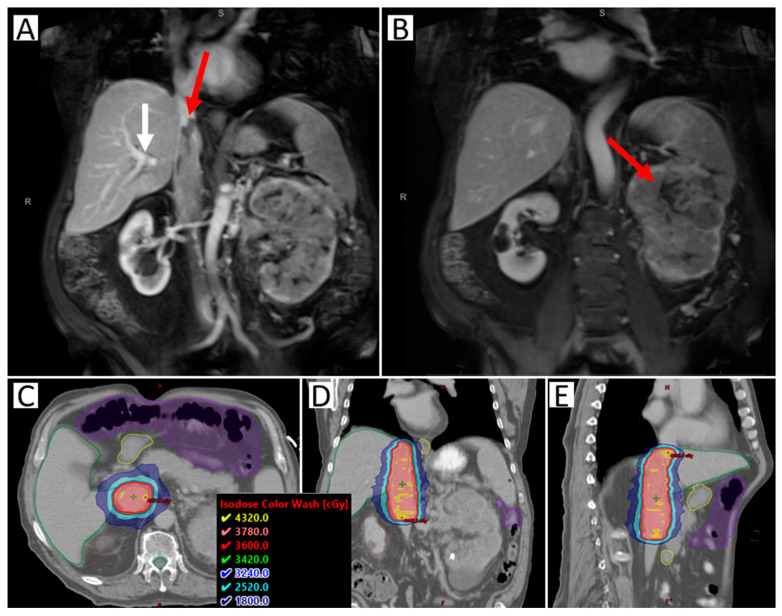
Images of a sample case of a patient with RCC IVC tumor thrombus level III, making resection not possible. The patient was treated with SAbR 36 Gy/3 fractions. (**A**) Coronal abdominal MR with contrast; red arrow highlighting the superior extent of the tumor thrombus; white arrow highlighting tumor thrombus extension beyond the branching of the hepatic artery (blockage the leads to Budd-Chiari Syndrome, making surgery more complicated). (**B**) Coronal abdominal MR with contrast; right arrow highlighting the large left primary kidney tumor. (**C**–**E**) Axial, coronal, and sagittal CT with radiation dose distribution shown. Nearby organ at risk highlighted include liver (green), duodenum (yellow), bowel space (purple), and spinal cord (green).

**Figure 2 cancers-14-04693-f002:**
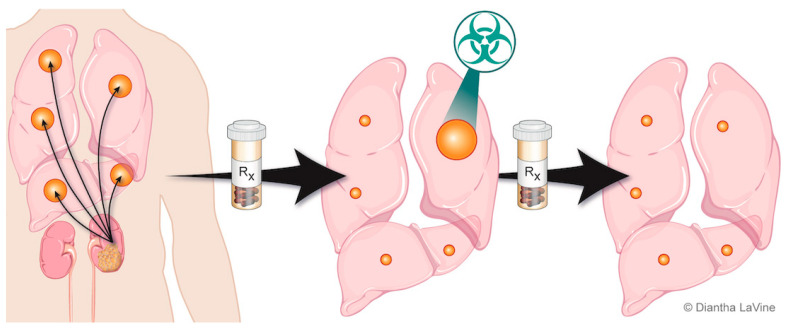
Depiction of oligoprogressive metastatic RCC. The metastatic disease is treated with systemic therapy and has a favorable response except at limited site(s). The limited progressive disease can be targeted with radiation therapy (SAbR), which provides durable control so the patient can remain on the same systemic therapy.

**Table 1 cancers-14-04693-t001:** Summary of key data supporting SAbR for RCC in various clinical scenarios.

Reference	Patients	Study Design	Follow-Up	Key Findings	Notes
**Primary RCC**					
Ponsky, Radiotherapy and Oncology 2015 [26]	19	Phase 1 dose escalation	13.7 months (median)	48Gy/4Fx without dose-limiting tox, acute G4 tox 5.2%, 80% stable disease, 20% partial response	
Sun, American Journal of Roentgenology 2016 [25]	40	Phase 1 dose escalation	1.5 years (mean)	LC 92.7%, mean tumor volume growth rate decreased from 21.2 cm^3^/y prior to SAbR to −5.35 cm^3^/y post-treatment	No statistically significant change in tumor enhancement post radiation
Siva, Cancer 2017 [28]	223	Pooled Analysis of individual patient data pooled	2.6 years (median)	2-yr LC 97.8%, CSS 95.7%, PFS 77.4%. 4-yr 97.8%, 91.9%, and 65.4%. G3+ tox 1.3%	No difference in LF between single Fx cohort and multi-fraction cohort
Correa, European Urology Focus 2019 [30]	372	Meta-analysis	28 months (median)	Random-effect estimates LC 97.2%, G3+ tox 1.5%	hisology confirmed 78.9%
Siva, IJROBP 2020 [31]	95	Pooled Analysis of individual patient data pooled	2.7 years	2-yr CSS 96.1%, OS 83.7%, PFS 81.0%. 4-yr LF 2.9%, DF 11.1%, any failure 12.1%. G3+ tox 0%	T1b tumors only
**Locally Advanced RCC**					
Hannan, Cancer Biology Therapy 2015 [32]	2	Case series, neo-SAbR for RCC IVC TT followed by RN and thrombectomy	20 months	24mo survival; other patient survived 18 months post-SAbR. No acute or late treatment-related toxicity.	Comparable to reported median survival of 20mo in patients with level IV IVC-TT treated with surgical resection
Margulis, IJROBP 2021 [35]	6	Prospective Phase 1-2 Neo-SAbR for RCC IVC TT followed by RN and thrombectomy	24 months	G3+ AEs after within 90s of surgery 4%, Of 3 patients with mets at Dx, 1 CR and 1 partial abscopal response without use of concurrent systemic therapy	
**Oligometastatic RCC**					
Svedman, Acta Oncologica 2006 [20]	30	Prospective Phase 2 trial	52 months median	LC 79% (loss to follow-up) or 98% if all stable, OS 32 months	
Zhang, IJROBP 2019 [42]	47	Retrospective review	30 months (median)	2-yr LC 91.5%, G3+ tox 0%, median freedom from systemic therapy 15.2 months, 2-yr OS 84.8%	
Tang, Lancet Oncology 2021 [43]	30	Single arm phase 2	17.5 months	Median PFS 22.7mo, 1-yr systemic therapy-free survival probability 82%, 10% G3+ tox	
Hannan, European Urology Oncology 2022 [44]	23	Single arm phase 2	21.7 months	1-yr freedom from systemic therapy 91.3%, 1-yr PFS 82.6%, LC 100%, G3/4 tox 0%, one death due to immune-related colitis on checkpoint inhibitor therapy;	1-yr OS 95.7%, 1-yr CSS 100%
**Oligoprogressive RCC**					
Schoenhals, Advances in Radiation Oncology, 2021 [54]	36	Retrospective review	20.4 months (median)	Median PFS 9.2 months, median OS 43.4 months, 1-yr LC 93%, G1-2 tox 33%, 1 G5 tox related to treatment or disease progression	
Cheung, European Urology 2021 [61]	37	Prospective multicenter	11.8 months	1-yr LC 93%, median PFS 9.3mo, median time to change systemic therapy 12.6mo, no G3+ toxicities	
Hannan, European Urology Oncology 2021 [62]	20	Prospective phase 2	10.4 months	LC 100%, median time of new systemic therapy or death 11.1 months, median duration of SAbR-aided systemic therapy 24.4 months, G3+ tox 5%	

Legend: AE: adverse event, CR: complete response, CSS: cancer-specific survival, DF: distant failure, Dx: diagnosis, Fx: fractions, Gy: Gray, IJROBP: International Journal of Radiation Oncology*Biology*Physics, IVC: inferior vena cava, LC: local control, LF: local failure, RN: radical nephrectomy, PFS: progression-free survival, RCC: renal cell carcinoma, SAbR: stereotactic ablative radiation therapy, tox: toxicity, TT: tumor thrombus, yr: year.

## Data Availability

Not applicable.

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
