# Peer review of "The Emerging Role of Radiation Therapy in Renal Cell Carcinoma"

_cancers, 2022, doi:10.3390/cancers14194693_

Round 1

Reviewer 1 Report

Very interesting article ! The studied topic presents practical importance.  I agree with the authors that this is a growing field of interest as this is documented by the growing number of relevant studies in the literature. The paper is well written and correct from a methodological point of view. I do not have any additional comments

Author Response

Thank you.

Reviewer 2 Report

The aim of the paper is to discuss the rationale and available evi- 16 dence for the integration and sequencing of SAbR with local and systemic therapies for RCC. The manuscript is clear and presented in a well structured manner. Results are reported clearly and appropriate. Tables and figures properly show the data. The discussion is adequate with current citations. The conclusions are consistent with the evidence. The cited references are mostly within the last 5 years, no self-citations were found.

Author Response

Thank you.

Reviewer 3 Report

Excellent narrative review regarding the role of modern radiation therapy on different RCC clinical scenarios.

As the review is dense in clinical data, I would suggest to organize the most important clinical results in tables for quick consultation. That would help to reduce the data density in the text also. The tables should cover the different clinical scenarios (primary RCC, oligometastatic disease, oligoprogressive disease, and combined treatmets toxicity). One additional table with the ongoing trials would be useful.

(lines 41-43) "when delivered with a higher dose per-fraction, RCC has subsequently been shown to be radiosensitive in numerous in vivo and in vitro studies [11,12]."

The studies could be numerous but the references provided by the authors are sparse: only two. In fact, reference 12 is not related with hypofractionated radiation treatments, as they are used for SABR. Because hypofractioned radiotherapy is the cornstone of SABR treatments, further radiobiological support for this strategy is requested.

In reference 11 (Walsh et al.), some experiments were done in xenografts using tumor volume as endpoint. That experimental setting do not allow to define radiosensitivity but radioresponsiveness. 

As it is defined in a reference Radiobiology book (Basic Clinical Radiobiology), cellular radiosensitivity parameters are derived from the cell survival curve but not from tumor growth delay assays as the ones mentioned by Walsh et al. 

Therefore, the term "radiosensitive" should be replaced by "radioresponsive" and, if possible, more bibliographic support on radiobiological studies should be provided. 

Author Response

Thank you for your feedback.

  • We have included a table that summarizes the key studies for SAbR for RCC in various clinical scenarios.
  • We have made your suggested edit and used the term "radioresistant."
